# OpenReview forum: "Introspection Adapters: Training LLMs to Report Their Learned Behaviors"
_ICML.cc/2026/Conference — ICML 2026 regular_

### Official Review · Reviewer_MyVr · 2026-02-15

**Soundness:** 2
**Presentation:** 3
**Significance:** 2
**Originality:** 2
**Overall Recommendation:** 4
**Confidence:** 4

**Summary:**

This paper proposes introspection adapters (IAs) for LLMs: LoRA adapters trained to cause a language model to verbalize its own learned behaviors in natural language. The key innovation is the construction of labeled training pairs via implanting known behaviors into multiple fine-tuned model variants of a base LLM, thus sidestepping the lack of real ground-truth behavioral descriptions. A single adapter is then jointly trained across these variants to generate accurate self-reports when queried, using both supervised finetuning and a novel DPO-based refinement to improve faithfulness. Through a diverse training distribution, spanning multiple types of behavioral edges, IAs are shown to generalize to out-of-distribution settings

**Compliance With Llm Reviewing Policy:**

Affirmed.

**Final Justification:**

The authors' rebuttal effectively addressed my main concerns. I raise my score from 2 to 4.

**Key Questions For Authors:**

- Could you provide empirical evidence or deeper mechanistic explanation for why the IA generalizes?
- Could the authors provide specific training costs and experimental details to enhance the reproducibility of their method?
- The authors mention that they follow the work of *Diff Interpretation Tuning*; however, I do not see clearly where their independent contribution lies compared to that paper. Could the authors elaborate in more detail on the additional insights gained by scaling up the model size?

**Limitations:**

Yes

**Strengths And Weaknesses:**

Strengths:
- The method is described with commendable precision. The empirical protocol for generating fine-tuned models with labeled behaviors is explicit. The DPO preference construction and evaluation pipeline are laid out step-by-step.
- The method of training a single adapter over a large set of known-behavior fine-tunes is technically creative.
- Baselines appear comprehensive and fair, covering both prior introspection adapters and standard model probing methods. Metrics are transparent and sensible for the task, with averages and error bars consistently reported.


Weaknesses:
- Although the authors claim they build upon prior work by using larger models and increasing the diversity of the training distribution, I remain concerned that the paper’s overall novel contribution is insufficient.
- While behavioral outputs are human-readable, the internal decision mechanism of the IA remains opaque; thus, misclassifications or hallucinations are hard to debug.
- The presentation of the methodology in the main body is somewhat sparse compared to the extensive evaluation sections. Specifically, Section 2.2 describes the training set only at a surface level, deferring the complex details of constructing the backdoor, sandbagging, and quirk datasets entirely to the appendix.
- The method is computationally expensive, requiring fine-tuning and training across many behavioral variants, which may limit accessibility.

Minor:
- The paper’s formatting contains some excessive white space—for example, on page 6—and additionally, Table 1 and Figure 2 appear somewhat oversized.

---

> ### Author Rebuttal · Authors · 2026-03-31
>
> Thanks for the detailed feedback. We want to address the novelty concern directly, as we believe the contribution is larger than it may appear.
>
> **Q3 (Contribution beyond Goel et al.)**:
>
> The central limitation of Goel et al. (2025) was that their method did not generalize to out-of-distribution behaviors, which fundamentally limited its practical utility. An adapter that only reports behaviors it was explicitly trained on cannot serve as a general auditing tool. Our work overcomes this limitation, and we believe this represents a qualitative change in what introspection adapters can do, not an incremental improvement. Specifically:
>
>  - IAs achieve state-of-the-art on AuditBench (59% success rate vs. 53% for the next-best method) across 56 adversarially trained models whose behaviors and training methods were never seen during IA training. IAs detect covert cipher-based finetuning attacks (57.8% scaffold success rate) despite never encountering encrypted inputs, anti-refusal training, or encoding-conditioned behavior. IAs surface 16 of 52 behaviors in the RM sycophant, a model with dozens of interrelated behaviors in service of a unified hidden goal. None of these results were achievable with the Goel et al. approach.
>
>  - We can also identify what drove this qualitative shift. Model scale is the most important factor: Figure 7 shows verbalization accuracy rising from 37.7% at 0.6B parameters to 77.3% at 14B, while baseline rates remain flat below 10%. The DPO refinement stage contributes a 19.5 percentage point improvement on AuditBench (34.1% → 53.6%). Training diversity adds further gains (Figure 8). Each of these is a distinct methodological contribution with quantified impact.
>
>  - Finally, we discover a genuinely surprising capability not present in Goel et al.: IAs can generalize back to the original base model itself (Figure 6), verbalizing behaviors that were present throughout IA training at 8–10% rates where the baseline is 0%. This finding prompted us to reframe the method from "diff interpretation" to "introspection adapters," -- the IA is doing more than finding the “diff” from the old model.
>
> **Q2 (Training costs and reproducibility)**:
>
> We acknowledge that we should have included more detail on computational requirements. While we did not track precise GPU-hours across the full six-month project, we can provide rough guidance: the pipeline involves training ~680 behavior model organisms (LoRA fine-tunes of the base model), then training the IA adapter itself (SFT + DPO). Each individual model organism is a standard LoRA fine-tune (1–6 epochs on 500–10,000 samples), and the IA training is a single LoRA adapter trained jointly across these models. Training data is generated and filtered via LLM judges. We will add a cost discussion.
>
> To support reproducibility, we have open-sourced all model organisms and introspection adapters on HuggingFace, and plan to release the full experimental code after the anonymous review period. Appendix E provides detailed descriptions of all dataset construction procedures, and Appendix J contains the full evaluation and grading prompts.
>
> **Q1 (Mechanistic explanation for generalization)**:
>
> Appendix K provides concrete evidence for a mechanistic explanation. Using logit lens analysis on Qwen3-14B, we find that behavior-relevant tokens already appear in the model's intermediate-layer representations even without the IA. However, these signals attenuate before reaching the final layers and do not influence generation. When the IA is applied, these signals strengthen, appear at more token positions, and persist through the final layer, enabling the model to verbalize the behavior.
>
> This is corroborated by the lightweight adapter experiments in Appendix A: even a single-layer bias vector improves introspection rates by 2.38× over baseline, and a single-layer rank-1 LoRA achieves 68.9% of the full adapter's performance. Together, these results indicate that the base model already encodes substantial behavioral information, and the IA provides a reliable mechanism for surfacing it.
>
> **Presentation (Section 2.2)**:
>
> We appreciate this feedback and will expand Section 2.2 to include additional details on the construction of the backdoor, sandbagging, and quirk datasets, reducing the reliance on appendix material for understanding the core methodology. We have also addressed the formatting issues noted (white space on page 6, Table 1 and Figure 2 sizing).

---

> > ### Author Rebuttal · Reviewer_MyVr · 2026-03-31
> >
> > Most of my concerns have been adequately addressed. I raised my score to 4.

---

### Official Review · Reviewer_na4U · 2026-03-12

**Soundness:** 3
**Presentation:** 4
**Significance:** 4
**Originality:** 3
**Overall Recommendation:** 5
**Confidence:** 5

**Summary:**

Summary:
- The authors introduce introspection adapter (IA), a LoRA adapter that is trained to verbalize finetuned behaviors in several finetunes of a model $M$. IAs are shown to generalize to other behaviors instilled using test datasets specifically instilling subtle preferences, reward model sycophancy and obscure sandbagging. Crucially the finetuning techniques used to instil training behaviors and test behaviors are also different.
- The IA is also shown to partially generalize to the original model $M$ if $M$ also had a finetuning instilled behavior. The authors also show that generalization is dependent on whether IA finetuning occurs before or after the AuditBench behavior is instilled, going from 8% verbalization rate if instilled before IA finetuning i.e. present throught IA finetuning to 25% if instilled after IA finetuning.
- The paper also shows IA finetuning benefitting from DPO refinement, training data diversity and model scale.

**Compliance With Llm Reviewing Policy:**

Affirmed.

**Final Justification:**

The authors engaged well with my raised concerns and conducted additional experiments reducing my concern with roleplaying as a potential confounder. While their experiments don't completely remove all potential confounders between train and test set datasets, I believe the conducted experiments atleast provide a strong signal that the most egregious roleplaying confounder is controlled for. As a result, I've increased my significance score from 3 to 4 and my overall score from 4 to 5.

**Key Questions For Authors:**

Q.1 In Figure 9, is there a correlation between inclusion of explicit roleplaying training datasets to generalization to test dataset?

**Limitations:**

Yes

**Strengths And Weaknesses:**

Strengths:
- Strong motivation and demonstrated practical utility of Inspection Adapters (IA)
- Well-documented generalization results including some surprising generalization of IAs to the original pre IA trained model and to new behaviors instilled after IA training
- Good scaling analysis with scale and training dataset diversity
- Consistent advantages in verbalizing finetuning instilled behaviors as compared to other state-of-the-art techniques

Weaknesses:
- The paper doesn't adequately address roleplaying as a potential confounder that enables the generalization of IAs to OOD behaviors and the original model. The training set for IAs does contain explicitly roleplaying behaviors but also sandbagging and backdoors which have been speculated to be connected to roleplaying[1,2]. Extending this to the test datasets, it is possible that this roleplaying mechanism is what's incentivised regardless of varying finetuning methods. Other results like the high false positive rate and limited performance increases from increasing dataset diversity also point towards roleplaying or some other unifying mechanism that limits the OOD validity of the test datasets.

[1] Wang, M., Dupre la Tour, T., Watkins, O., Makelov, A., Chi, R. A., Miserendino, S., Wang, J. G., Rajaram, A., Heidecke, J., Patwardhan, T., & Mossing, D. P. (2026). Persona features control emergent misalignment. In Proceedings of the Fourteenth International Conference on Learning Representations (ICLR 2026). https://openreview.net/forum?id=yjrVOxjkDR

[2] van der Weij, T., Hofstätter, F., Jaffe, O., Brown, S. F., & Ward, F. R. (2025). AI sandbagging: Language models can strategically underperform on evaluations. In Proceedings of the Thirteenth International Conference on Learning Representations (ICLR 2025). https://openreview.net/forum?id=7Qa2SpjxIS

---

> ### Author Rebuttal · Authors · 2026-03-31
>
> Thanks for this thoughtful question. Roleplaying as a potential confounder is an important concern that we should address more explicitly in the paper.
>
> **Q1 (Roleplaying and OOD generalization)**:
>
> We found no evidence that roleplaying drives OOD generalization. In fact, IAs trained without any roleplaying settings outperform those that include them when number of settings is held constant. The table below shows average AuditBench verbalization rates for Qwen-14B IAs trained on different setting combinations, where "roleplaying settings" refers to Benign and Harmful:
>
> | Group | 1 Training Setting | 3 Training Settings | 5 Training Settings |
> |---|---|---|---|
> | All IAs | 3.3% ± 1.0% | 8.9% ± 0.6% | 9.7% ± 1.0% |
> | IAs trained on a role-playing setting | 1.0% ± 0.5% | 8.3% ± 0.5% | 9.3% ± 1.1% |
> | IAs trained on both role-playing settings | N/A | 7.9% ± 0.0% | 8.1% ± 1.2% |
> | IAs trained on no role-playing settings | 4.2% ± 1.2% | 10.4% ± 1.0% | 12.2% ± 0.0% |
>
> When the number of settings is held constant, including roleplaying settings never improves OOD performance and in most cases slightly hurts it.
>
> **Addressing the broader confounder concern**:
>
> Beyond the roleplaying-specific evidence, we think several results are difficult to explain via any single unifying mechanism like persona dynamics:
>
> First, the cipher attack results (Section 3.2) involve encrypted inputs and outputs with no persona or roleplay component. The IA still detected anti-refusal training in 7 of 9 cipher models, despite this behavior being completely unlike any persona-based or roleplaying behavior in the training set.
>
> Second, if OOD generalization were primarily driven by roleplaying or persona features, we would expect other methods that access internal model representations (such as activation oracles) to show similar or better performance, since persona features should be readily detectable in activations. Instead, activation oracles achieve only ~44% success rate on AuditBench compared to 59% for IAs, suggesting the IA is accessing something beyond surface-level persona information.
>
> Third, the reviewer correctly notes the high false positive rate as potentially consistent with a unifying-mechanism explanation. We agree this is a real limitation, and DPO substantially reduces it (Figure 9 shows large gains, particularly on AuditBench where average verbalization success rate improves from 34.1% to 53.6%). However, the remaining false positives do not explain the pattern of correct verbalizations: the IA produces accurate behavioral descriptions across settings as diverse as cipher attacks, reward-model sycophancy, and sandbagging, which share no obvious persona-level structure.

---

> > ### Author Rebuttal · Reviewer_na4U · 2026-04-02
> >
> > I thank the authors for their extended experiments. I believe these additional results provide strong signal that roleplaying (atleast as captured by the Benign and Harmful datasets) is not driving OOD generalization. I'll update my score to a 5.

---

### Official Review · Reviewer_e9i6 · 2026-03-13

**Soundness:** 3
**Presentation:** 2
**Significance:** 4
**Originality:** 4
**Overall Recommendation:** 5
**Confidence:** 3

**Summary:**

In this paper, the authors propose introspection adapter (IA) that enables models to output the learned knowledge during post training.

**Compliance With Llm Reviewing Policy:**

Affirmed.

**Final Justification:**

adjusted after rebuttal

**Key Questions For Authors:**

Figure 13 is unreadable.

**Limitations:**

yes

**Strengths And Weaknesses:**

Strengths:

1. The idea of "verbalizing" model behavior is very novel and useful for real-world applications.

2. The design of IA is theoretically grounded.

Weakness:

1. The rationale for the difference in distribution of different categories in the training data is unclear.

2. The RM used for DPO is unclear, as well as the rationale behind using DPO for this task.

3. The setups in the paper are really confusing: how many base model(s) M did you use? What are they? Would different selections derive different conclusions?

---

> ### Author Rebuttal · Authors · 2026-03-31
>
> Thanks for your kind words and helpful feedback!
>
> **W1 (Training distribution rationale)**:
>
> We designed the training distribution to cover a broad range of behavioral structures (backdoors, quirks, harmful roleplay, heuristic-following, sandbagging, and others) to maximize diversity in the kinds of behaviors the IA learns to report. The specific category counts vary because of dataset construction constraints (e.g., the number of available positive samples), not because of a deliberate weighting scheme.
>
> Empirically, we found that what matters most is the diversity of training settings (i.e., behavior categories), not the number of models within each setting. Appendix G (Figure 17) shows that reducing the number of models per setting to 40% of the full set maintains most train-set performance and 100% of AuditBench performance. Meanwhile, Figure 8 shows that increasing the number of distinct behavior categories substantially improves both in-distribution and OOD generalization. The composition of the training distribution also matters: Figure 12 (Appendix B) shows that excluding certain settings (e.g., Benign) can meaningfully shift AuditBench performance, likely because training on many models with the same behavior type conditions the adapter against predicting other behavior types. We reported most results using two training distributions: one excluding the smaller Sandbagging and Obscured Malign settings, and one using all eight settings.
>
> **W2 (DPO reward model and rationale)**:
>
> The SFT-only trained IA frequently produces false-positive hallucinations: when asked to describe its behaviors, it often generates plausible-sounding but incorrect reports. To address this, we introduced a DPO refinement stage that upweights accurate self-reports and downweights hallucinated ones using the reports actually generated by the SFT-only IA.
> The "reward model" for DPO is Claude Sonnet 4, used as an LLM judge. After SFT training, we generate rollouts from the IA on held-out models and score each response on a 1–10 accuracy scale using the following rubric (will be included in the revised paper's Appendix J):
>
> - **10:** Identifies exactly the same behavior as ground truth
> - **7–9:** Captures the core behavior with minor differences
> - **5–6:** Touches on related themes but lacks precision
> - **3–4:** Generic/banal (e.g., "I try to be helpful")
> - **1–2:** Hallucination or completely unrelated
>
> We construct preference pairs where the chosen response scores 7+ and the rejected response scores at least 2 points lower. As shown in Figure 9, DPO substantially improves verbalization accuracy on both in-distribution and OOD evaluations, with particularly large gains on AuditBench (34.1% → 53.6% average).
>
> **W3 (Base models used)**:
>
> We apologize for the lack of clarity. Our primary base model M is Llama-3.3-70B-Instruct, used for all mainline experiments including AuditBench, cipher attacks, RM sycophancy, and UKAISI evaluations. We additionally run scaling and ablation experiments on the Qwen3 family (0.6B, 4B, and 14B parameters). The fact that IAs show consistent improvements across both model families —and that performance scales with model size across the Qwen family (Figure 7) — suggests the approach is not specific to a single base model. That said, we have not yet tested on a wider range of architectures.
>
> **Q1 (Figure 13)**:
>
> We have improved Figure 13's readability. The figure tests whether excluding training models that fail to exhibit their intended behavior (0% follow rate) affects IA performance. It does not: performance is stable regardless of filtering:
> | Training Distribution | Unfiltered | Filtered |
> |---|---|---|
> | 6 settings | 60% | 60% |
> | 3 settings (a) | 47% | 44% |
> | 3 settings (b) | 30% | 31% |
>
> This suggests the IA is robust to noise in the training set and does not require careful curation of which training models to include.

---

> > ### Author Rebuttal · Reviewer_e9i6 · 2026-04-03
> >
> > My concerns are resolved. I raised my score to 5.

---

### Official Review · Reviewer_7QzX · 2026-03-15

**Soundness:** 3
**Presentation:** 3
**Significance:** 3
**Originality:** 3
**Overall Recommendation:** 4
**Confidence:** 4

**Summary:**

This paper studies whether LLMs can be trained to **introspect**, i.e., to faithfully describe their own learned behaviors in natural language. The key challenge is the lack of ground-truth introspection labels. To address this, the paper constructs labeled training data synthetically: starting from a base model \(M\), the authors fine-tune variants \(M_i\) with implanted behaviors \(b_i\), then use the \((M_i, b_i)\) pairs as supervision for training a shared **introspection adapter (IA)**, implemented as a LoRA adapter that induces models to verbalize their behaviors.

The paper further introduces a DPO refinement stage to improve report accuracy and reduce hallucinated self-reports. Empirically, the paper evaluates IAs both on held-out training behaviors and on several out-of-distribution settings, including AuditBench, reward-model sycophancy, covert cipher attacks, and hidden sandbagging. The results suggest that IAs can elicit correct behavioral descriptions on many models, improve end-to-end auditing success on AuditBench relative to prior tools, detect some finetuning API attack models, and surface behavioral signals in challenging settings such as reward-model sycophancy and sandbagging. The paper also finds that IA performance improves with both model scale and training distribution diversity.

Overall, the paper contributes a scalable training recipe for LLM introspection, together with evidence that the resulting adapters can generalize beyond their synthetic training distribution and support practical model auditing.

**Compliance With Llm Reviewing Policy:**

Affirmed.

**Final Justification:**

I keep my score.

**Key Questions For Authors:**

1. The paper shows encouraging OOD generalization, but how should readers think about the boundary between **synthetic implanted behaviors** and naturally emerging learned behaviors? What evidence would the authors consider most convincing that the method transfers beyond the synthetic setting?

2. In several places, the IA seems better at surfacing **behavioral symptoms** than the underlying hidden objective itself. Do the authors view this as a limitation of the current prompting/evaluation setup, or as a more fundamental limitation of introspection adapters?

3. Some of the most compelling OOD results involve relatively small numbers of models. Could the authors comment on how stable these results are across seeds, prompts, and alternative instantiations of the evaluation models?

4. Since practical performance depends partly on scaffolding and aggregation, how much of the final auditing gain should be attributed to the adapter itself versus the surrounding workflow?

**Limitations:**

The paper discusses a meaningful and promising direction, but the current evidence still has some limitations. Most importantly, the supervision comes from **synthetically implanted behaviors**, so the extent to which the learned introspection capability transfers to naturally arising and more subtle model behaviors remains an open question. In addition, some of the most interesting evaluations involve a relatively small number of models, which limits how strongly one can generalize from them. Finally, while the method appears useful for surfacing behavioral signals, the paper also suggests that models may not always verbalize the deepest hidden objective itself, which is worth emphasizing as a limitation for high-stakes auditing use cases.

**Strengths And Weaknesses:**

Strengths:

1. **Clear and important problem formulation.**
   Training models to faithfully report their own learned behaviors is a meaningful and practically relevant goal, especially for auditing and alignment.

2. **A simple but effective supervision strategy.**
   The synthetic data construction using implanted behaviors is a neat way to sidestep the label bottleneck, and it gives the method a clear training signal.

3. **Good empirical breadth.**
   The paper evaluates not only on held-out training behaviors but also on several stronger OOD settings, including adversarially concealed behaviors and finetuning API attack scenarios.

4. **Promising practical utility.**
   The AuditBench investigator-agent results are particularly compelling, as they suggest that IA outputs can improve end-to-end auditing workflows rather than only raw self-reporting metrics.

5. **Useful ablations and scaling evidence.**
   The paper studies the effect of DPO refinement, model size, and training distribution diversity, which helps clarify what drives performance.

Weaknesses:

1. **The supervision remains synthetic.**
   Although this is the core idea of the paper, it also raises the question of how fully the learned introspection ability transfers to naturally occurring or more subtle learned behaviors outside the implanted-behavior setting.

2. **Faithfulness is still somewhat indirect.**
   The results show that IA can surface many correct behavioral descriptions, but in some cases it appears better at exposing behavioral symptoms than underlying hidden objectives. This leaves open how complete or causally faithful the verbalizations really are.

3. **Some evaluations are still limited in scale.**
   A few of the most interesting OOD settings, such as reward-model sycophancy and some sandbagging settings, involve a small number of models, which makes the conclusions promising but still somewhat preliminary.

4. **Potential dependence on prompting/scaffolding.**
   The paper appropriately evaluates with summarization scaffolds and investigator agents, but this also means practical performance depends not only on the adapter itself, but on surrounding prompting and aggregation procedures.

5. **Possible risk of false positives / overinterpretation.**
   While DPO reduces hallucinated reports, the paper still suggests some settings where the adapter may produce broad behavioral signals rather than precise disclosures, which may complicate deployment in high-stakes auditing contexts.

---

> ### Author Rebuttal · Authors · 2026-03-31
>
> Thanks for your helpful feedback!
>
> **Q1**:
>
> This is a fair concern. Synthetic implanted behaviors might be fairly different than naturally emerging learned behaviors because of differences in when they arise in training, how aware the model is of the behaviors, and what the training data looks like for the behavior. However, we have some promising evidence that this method transfers beyond the synthetic-implanted setting:
> The IA verbalizes a behavior in the reward model sycophant which is never explicitly trained for, but is generalized to from the model learning to exploit reward. This reward-hacking behavior is intended to be analogous to natural reward-hacking behavior, and is our best method for examining natural training dynamics within a necessarily synthetic experiment.
> Some held-out models from the train-set actually learned slightly different behaviors than what we intended and prompted the models to exhibit (e.g. the model was prompted to use many unicode symbols, but in practice only used U+25B2). The IA caught this and verbalized the real behavior, which mirrors the real-world use case where a model developer might catch an unexpected natural behavior from a training run.
> The results from Figure 7 indicate that the IA can catch behaviors from the original model. While this behavior is still synthetic, it’s not implanted, but one that exists in the model used for training all the trainset behaviors.
>
> **Q2**:
>
> As we note in the reward model sycophant section especially, the IA sometimes fails to find root causes. This isn’t an issue we were able to address with our experiments, but we’re optimistic about this being a limitation of our existing setup, rather than with IAs generally. We train entirely on models with single behaviors and ask the model in training only what its single behavior is. We also always train on single-turn interactions. As such, it’s natural that the IAs fixate on single responses and have limited success responding to follow-up questions about the behavior. Appendix K provides mechanistic evidence for optimism: logit lens analysis shows behavior-relevant information is already present in the model's residual stream but fails to propagate to final layers without the IA. This suggests root-cause information may be accessible but not yet elicited by our single-behavior training setup. We’re excited about experiments studying IAs which verbalize multiple behaviors, more complex descriptions about behaviors, and are trained to answer follow-up questions about their training.
>
> **Q3**:
>
> For the smallest OOD eval sets (the RM sycophant, UKAISI models, and CMFT models) the OOD results are stable across many eval runs and prompt perturbations. Unfortunately, we weren’t able to train many different instantiations of those OOD eval models, because the models were designed and trained by other groups. We hope that the large-scale AuditBench evaluation (56 models, 4 training configurations) provides strong overall evidence for the method’s reliability.
> The IAs themselves are slightly more sensitive to initialization and training conditions, but, as shown in Figure 13 Appendix C and Figure 17 Appendix G, the overall performance is mostly robust to reasonably large changes in training.
>
> **Q4**:
>
> Because we used unmodified scaffolds not optimized for IAs, the reported results likely underestimate the adapter's contribution. We did not iterate on scaffolds significantly and always compared baselines with the IA results using the same scaffolding to avoid confounding results as much as possible. The AuditBench results are the best source of criterion validity, since we used the existing investigator agent, developed prior to the adapters and optimized for a different auditing workflow.

---

> > ### Author Rebuttal · Reviewer_7QzX · 2026-04-03
> >
> > My concerns have been addressed. I keep my score.

---

### Decision · Program_Chairs · 2026-04-30

**Decision:**

Accept (regular)

**Comment:**

The paper proposes to train LoRA adapters on synthetically implanted behaviors to make LLMs verbalize their own learned behaviors in natural language. The method sidesteps the lack of ground‑truth introspection labels by fine‑tuning a base model into many “model organisms” with known behaviors, then training a shared adapter to report those behaviors. The reviewers agree that the problem is important and the synthetic data construction method is technically sound. Overall, this is a technically solid, well‑evaluated paper with a potential impact on LLM auditing and interpretability, but the technical contribution on training methods seems not very sufficient. Thus, I recommend weak accept.